# Role of Melatonin in Daily Variations of Plasma Insulin Level and Pancreatic Clock Gene Expression in Chick Exposed to Monochromatic Light

**DOI:** 10.3390/ijms24032368

**Published:** 2023-01-25

**Authors:** Chao Song, Zixu Wang, Jing Cao, Yulan Dong, Yaoxing Chen

**Affiliations:** 1College of Veterinary Medicine, China Agricultural University, Haidian, Beijing 100193, China; 2Department of Nutrition and Health, China Agricultural University, Haidian, Beijing 100193, China

**Keywords:** monochromatic light, clock gene, insulin, melatonin, pancreas

## Abstract

To clarify the effect of monochromatic light on circadian rhythms of plasma insulin level and pancreatic clock gene expression and its mechanism, 216 newly hatched chicks were divided into three groups (intact, sham operation and pinealectomy) and were raised under white (WL), red (RL), green (GL) or blue (BL) light for 21 days. Their plasma and pancreas were sampled at six four-hour intervals. For circadian rhythm analysis, measurements of plasma melatonin, insulin, and clock gene expression (*cClock*, *cBmal1*, *cBmal2*, *cCry1*, *cCry2*, *cPer2*, and *cPer3*) were made. Plasma melatonin, insulin, and the pancreatic clock gene all expressed rhythmically in the presence of monochromatic light. Red light reduced the mesor and amplitude of plasma melatonin in comparison to green light. The mesor and amplitude of the pancreatic clock gene in chickens exposed to red light were dramatically reduced, which is consistent with the drop in plasma melatonin levels. Red light, on the other hand, clearly raised the level of plasma insulin via raising the expression of *cVamp2*, but not *cInsulin*. After the pineal gland was removed, the circadian expressions of plasma melatonin and pancreatic clock gene were significantly reduced, but the plasma insulin level and the pancreatic *cVamp2* expression were obviously increased, resulting in the disappearance of differences in insulin level and *cVamp2* expression in the monochromatic light groups. Therefore, we hypothesize that melatonin may be crucial in the effect of monochromatic light on the circadian rhythm of plasma insulin level by influencing the expression of clock gene in chicken pancreas.

## 1. Introduction

The avian species circadian timing system consists of a central brain clock, and peripheral clock in other brain regions and tissues throughout the body, including muscle, adipose tissue, liver and pancreas [1]. The pineal, retina, and the suprachiasmatic nucleus (SCN) of the hypothalamus are among the central oscillators in birds [2,3]. All three pacemakers could entrain the peripheral oscillator and locomotor rhythm, the SCN receives a direct projection from the retina, via which environmental light synchronizes the approximately 24 h rhythm of the SCN with the exact 24 h rhythm of the environment [4,5]. The entrained timing signal from the SCN is forwarded via neural and hormonal signals and body temperature to the peripheral clocks. As in mammals, circadian oscillations in birds originate at the genetic level, the circadian oscillator is driven by the core circadian-feedback loop that consists of positive and negative genes [6,7]. The molecular basis of oscillators in birds consists of three positive clock genes, *cClock* (Circadian locomotor output cycles kaput), *cBmal1*, and *cBmal2* (Brain and muscle Arnt-like 1/2), as well as four negative clock genes, *cCry1*, *cCry2* (Cryptochrome 1/2), *cPer2* and *cPer3* (Period 2/3) [8], the BMAL-CLOCK heterodimer binds to E-box enhancer elements presented in the promoter of target genes, thereby activating the expression of negative genes, such as *Cryptochrome* (*Crys*) and *Period* (*Pers*).

The circadian clock drives cycles of energy storage and utilization in plants, flies, and mammals in anticipation of changes in the external environment imposed by the rising and setting of the sun [9]. One of the most noticeable rhythmic aspects of physiology is the daily variation of glucose tolerance and insulin sensitivity over the course of a 24 h day [10]. Over the past decades, it has become apparent that all organs involved in glucose homeostasis contain a functional clock [11], and studies with organ-specific clock-gene knockouts show that a functional clock is crucial for maintaining normal glucose homeostasis. Whole-body loss of clock function (e.g., in Clock or Bmal1 knockout animals) leads to hyperglycemia, glucose intolerance, and ultimately obesity and metabolic syndrome [12]. Fasting-induced hypoglycemia is caused by impaired glycogenesis and decreased hepatic glucose production when the liver clock is disrupted, whereas diabetes is caused by β-cell failure when the pancreas clock is disrupted [13]. The pancreas is the master regulator of glucose homeostasis, and the presence of an autonomous circadian pancreatic clock has been demonstrated not only in rodents [14], but also in human islets and dispersed human islet cells (that is, the cells were cultured as separator single cells, not as an intact islet) [15]. Pancreatic islets isolated from rats show a circadian rhythm in insulin secretion [16]. CLOCK and BMAL1 activate the transcription of genes involved in glucose-stimulated insulin secretion. Consistent with this observation, disruption of the pancreatic clock causes defective insulin secretion [16,17]. However, it is not clear whether there is a clock oscillator regulating insulin secretion in the chicken pancreas.

Diurnal rhythms in metabolism are driven not only by the circadian system but also by environmental and behavioral factors, including light [18], sleep [19], food intake [20,21], and physical activity. Increasing evidence suggests that when these external rhythms are out-of-sync with endogenous circadian rhythms, such as through exposure to bright light at night, sleeping during the daytime, or eating at night, the rhythmic insulin concentration in plasma is impaired. While light duration is considered the main factor in providing light information, changes in light wavelengths can also affect avian physiology and behavior. For example, our previous studies found that monochromatic lights could change the clock gene expression of chick central oscillators, including the pineal gland, the hypothalamus and the retina [22,23,24], and also demonstrated melatonin-mediated effects of monochromatic light on peripheral oscillator liver clock genes [25]. Green and blue monochromatic light combination therapy reduces oxidative stress and enhances B-lymphocyte proliferation through promoting melatonin secretion [26]. These findings implied that monochromatic light might play a critical role in the biological clock and developmental activity in the chicks. However, how monochromatic light affects the expression of the clock gene in pancreatic tissue and the secretion of insulin has been reported less frequently.

## 2. Materials and Methods

### 2.1. Animals and Treatments

All experiments were approved by the Animal Welfare Committee of the Agricultural Research Organization, China Agricultural University (Approval No. 20171114-2). A total of 144 newly hatched male chicks (Arbor Acres Broilers), which were purchased from Beijing Huadu Breeding Co., Ltd. (Beijing, China), were raised in white, red, green and blue light under an LED system (Zhongshan Junsheng Lighting Technology Co., Ltd., Zhongshan, China) for 21 days. Therefore, there were four light treatment groups: 400–700 nm white light group (WL), 660 nm red light group (RL), 560 nm green light group (GL) and 480 nm blue light group (BL). The light intensity was around 15 Lux, and the light regime was 12 light: 12 dark (light on at 08:30 as ZT 0 and off at 20:30 as ZT 12, zeitgeber time). The chickens had ad libitum access to feed and water, and the diets were formulated to meet the nutrient recommendations for poultry (NRC, 1994). The temperature in the chicken house was set at 32 °C for the first week and then reduced to 30 °C in the second week, and the relative humidity was maintained at 60% for the entire period.

The pinealectomy surgery (Pinx) was conducted at P3 as follows [27]: chicks were anaesthetized by intraperitoneal injection of pentobarbital sodium (30–40 μg/g body weight) and then were placed on a stereotaxic apparatus (ALCBIO, Shanghai, China). One mid-sagittal incision was made in the skin above the cranium, and a small portion of the skull was removed using a dental drill to expose the pineal gland. The meninges were cut away using microsurgical scissors, the pineal gland was removed with forceps, and the opening was packed with gelfoam to reduce bleeding. The skullcap was placed back on the bird, and the wound was then closed with surgical sutures and treated with a topical antibiotic ointment.

### 2.2. Sample

At the beginning of P21 of the light–dark illumination cycle, the light was off and chicks were maintained in constant dark conditions (DD).

The chicks were quickly sacrificed by decapitation under dim red light at 6 separate time points between CT 0 and CT 20 in 4 h intervals (CT 0, CT 4, CT 8, CT12, CT 16 and CT 20, circadian time). Blood samples were collected from veins and heparinized with 1000 UI/mL heparin in avian saline, followed by decapitation. After centrifugation at 3000× *g* for 15 min, the plasma was decanted and stored at −80 °C until melatonin and insulin measurement. The pancreas was removed, frozen in liquid nitrogen and stored at −80 °C (The head, middle and tail of the pancreas were collected at six time points).

### 2.3. Enzyme-Linked Immunosorbent Assay

Plasma melatonin and insulin were measured by ELISA (SLCY., Beijing, China). The operation was performed following the instructions provided in the kit. Each sample was tested in triplicate. The detection range of melatonin was 1 ng/L–80 ng/L, the detection range of insulin was 0.75 mU/L–30 mU/L, and the intra- and interassay coefficients of variations were <9% and <11%. The OD values were immediately measured by a microplate reader (Model 680, Bio-Rad, Hercules, CA, USA) at the 450 nm wavelength. The standard curve of the log melatonin and insulin concentration to OD value was determined, and sample melatonin and insulin was calculated according to this standard curve.

### 2.4. Real-Time Reverse Transcription Polymerase Chain Reaction (RT-PCR)

Following the manufacturer’s instructions, the pancreas RNA was isolated using an RNeasy Mini Kit (Qiagen, Germantown, MD, USA). The total RNA was extracted using TRIzol reagent (CW0580A, CoWin Biotech Co., Inc., Beijing, China). Spectrophotometric assessment of RNA quantity and integrity was conducted using a NanoDrop spectrophotometer (Thermo Fisher Scientific, Waltham, MA, USA). RNA was reverse transcribed into cDNA using a reverse transcription kit (R312-02, Vazyme Biotech Co., Ltd., Nanjing, China). RT-PCR amplification was performed using AceQ qPCR SYBR Green Master Mix (Q141-02; Vazyme Biotech Co., Ltd.). The experiments were repeated in triplicate, and the relative mRNA levels were normalized to the expression level of glyceraldehyde-3-phosphate dehydrogenase (GAPDH). The primers used are listed in Table 1.

### 2.5. Statistical Analysis

The data are presented as the mean ± standard error of the mean (SEM), the differences among daily time points were analyzed with one-way ANOVA using SPSS 25.0 software (SPSS, Chicago, IL, USA). Results were considered statistically significant when the *p* value was < 0.05. Correlation analysis, expressed as Pearson’s coefficient, was performed to determine the possible linear relationship between insulin and melatonin content in the plasma under different light colors, respectively.

We used a previously described method [28]. Briefly, we used the OriginPro 9.3 software (OriginLab Corp, Northampton, MA, USA) to determine whether the data had a 24 h circadian rhythm. First, we performed one way ANOVA on the data at different time points to determine significant differences between different time points. Then, we added the data into the formula: y = a + b × cos(x × pi/12 − c × pi/12), a, b and c represented the mesor, amplitude and acrophase, respectively. We could calculate the correlation coefficient of determination (R^2^) of the data under this formula and the values of the three parameters of mesor, amplitude, and acrophase. The significance of regression analysis determined at *p* < 0.05 was calculated using the number of samples, R^2^ and numbers of predictors (mesor, amplitude, and acrophase). If the *p*-value was less than 0.05, this dataset was considered to have had a circadian rhythm of 24 h.

The differences between the intact group and the pinealectomy group were analyzed by *t*-test using SPSS 25.0 software (SPSS, Chicago, IL, USA); *p*-values less than 0.05 were considered indicative of statistical significance.

## 3. Results

### 3.1. Effect of Monochromatic Light on Circadian Rhythms of Plasma Insulin Level, Plasma Melatonin Level and Pancreatic Insulin-Related Gene

To determine the effect of different wavelengths of light on the rhythmic secretion of pancreatic insulin, three-day-old chicks in the control, sham and pineal extraction groups were respectively raised under WL, RL, GL, and BL exposure for 3 weeks (Figure 1A). Plasma insulin levels in chicks exposed to monochromatic light displayed a day high and night low expression pattern and exhibited a circadian rhythm (see R^2^ and *p* value at Figure 1B). The insulin levels peaked at the middle of subjective day (CT 4), steadily decreased throughout the subjective day, and reached its lowest points right after the dark period (CT 16), before gradually rising. The content of plasma insulin concentrations and rhythmic parameters changed under monochromatic light. The plasma insulin levels were the highest in the RL group, which were significantly higher than that in the GL group (Figure 1F, *t*-test, *p* < 0.01). Compared to GL, RL elevated the mesor (+2.12 mU/L, +44.82%, *p* < 0.01) and amplitude (+2.7 mU/L, +126.2%, *p* < 0.01) of the plasma insulin (Figure 1J).

We identified the *cInsulin* and vesicle-associated membrane protein (*cVamp2)* mRNA expression in chicken pancreas. Under monochromatic light, the *cInsulin* gene expression was rhythmic (see R^2^ and *p* value at Figure 1D), showing a day low and night high expression pattern, peaking in the middle of subjective nights (Figure 1D), but the expression of the *cInsulin* (Figure 1H) gene and its rhythmic parameters (Figure 1L) were not altered. The *cVamp2* had a day high and night low expression pattern and had circadian rhythm (see R^2^ and *p* value at Figure 1G), with the highest expression under RL and the lowest expression under GL (Figure 1I). Compared to the GL group, the mesor and amplitude of the *cVamp2* were significantly higher in the RL group (Figure 1M).

In contrast, the melatonin levels increased gradually during the subjective day and peaked at the middle stage of subjective night (CT 16), then progressively decreased (Figure 1C), and plasma melatonin maintained diurnal variation and circadian rhythm (see R^2^ and *p* value at Figure 1C). In comparison to RL, GL increased the melatonin levels (Figure 1G), as well as the mesor (+11.29 pg/mL, +81.22%, *p* < 0.01) and amplitude (+7.52 pg/mL, +63.39%, *p* < 0.01) of the melatonin (Figure 1K). Under monochromatic light, plasma melatonin and insulin were found to be negatively correlated.

### 3.2. Effect of Pinealectomy on the Circadian Rhythms of the Plasma Melatonin Level under Monochromatic Light

We established a model of melatonin deficiency in broilers by pineal removal surgery. As shown in Figure 2A–D, the plasma melatonin levels decreased significantly after pineal gland removal at all time points (*t*-test, *p* < 0.05). Under monochromatic light, the mesor and amplitude of plasma melatonin were significantly reduced, but there was no significant change in the peak phase (Table 2). Under monochromatic light, plasma melatonin levels in the pinealectomy group decreased significantly when compared to the sham operation group (Figure 2E). After removing the pineal gland, the effect of monochromatic light on plasma melatonin levels (Figure 2F–G) and the rhythmic parameters (Figure 2H) such as the mesor, amplitude and acrophase were eliminated.

### 3.3. Effect of Pinealectomy on the Circadian Rhythms of the Insulin Level and Pancreatic Insulin-Related Gene under Monochromatic Light

To verify the effect of melatonin on insulin levels in plasma, we measured insulin levels in chick plasma after pineal gland removal. In contrast to the changes in melatonin, the insulin levels in the plasma increased significantly, particularly under GL and BL for all time points (Figure 3A, *t*-test, *p* < 0.05). Insulin level in plasma retained daily variation and circadian rhythm (see R^2^ and *p* value at Figure 3A). Compared with the sham operation group, the overall expression of insulin increased significantly under monochromatic light (Figure 3A).

Next, we examined the expression of insulin-related genes in pancreatic tissue after pineal gland removal. Under monochromatic light, the expression of *cInsulin* changed only at a few time points under monochromatic light (Figure 3B). The mesor value of *cInsulin* did not change significantly after pinealectomy, but the expression of amplitude was significantly decreased under WL, RL, and GL (Table 3 WL: −48.37%, RL: −39.51%, GL: −53.27%, *p* < 0.01). There was no significant change in the overall expression of *cInsulin* in the pinealectomy group compared to the sham operation group (Figure 3B). The effect of pineal gland excision on *cVamp2* expression and its rhythmic parameters was then discussed. As shown in Figure 3C, the expression of *cVamp2* increased significantly at majority of time points (*t*-test, *p* < 0.05), and *cVamp2* retained daily variations and circadian rhythms (see R^2^ and *p* value at Figure 3C). Similarly, we also compared the changes in overall expression of the *cVamp2*, and the results showed that the expression of the *cVamp2* in the pinealectomy group was significantly higher than that in the sham operation group under monochromatic light (Figure 3C).

### 3.4. Effect of Monochromatic Light on the Circadian Rhythms of the Pancreas Clock Gene Expression

When chicks were exposed to monochromatic light, pancreas clock gene expression changed significantly, although daily variations were retained (see Table 4 *p* value; Figure 4A–G). In addition to *cCry1*, GL enhanced the clock gene expression at nearly all time points when compared to RL (*p* < 0.05; Figure 4A–G). In terms of circadian rhythms, the pancreas clock gene expression was rhythmic (see R^2^ and *p* value at Figure 4), and their rhythm parameters changed notably under monochromatic light. Compared to WL, GL enhanced the mesor of all positive clock genes (Table 4, +20.72% to +34.38%, all *p* < 0.01) but did not alter their amplitude. In contrast, RL reduced the mesor of *cBmal2* (−19.82%, *p* < 0.01) and the amplitudes of *cBmal1* (Table 4, −28.0%, *p* < 0.01). Additionally, blue light had no effect on the mesor and amplitude of the positive clock gene. Monochromatic light delayed the acrophases of *cClock* (0.27 h–3.28 h, all *p* < 0.01), while advancing the acrophase of *cBmal1* under RL (−1.77 h) (Table 4; Figure 4A–C).

Consistent with the changes of the positive clock gene, GL increased the mesor of the negative clock gene, especially that of *cCry2* (+46.24%, *p* < 0.01) and *cPer3* (+51.47%, *p* < 0.01), and GL also increased the amplitudes of *cCry2* (+106.5%, *p* < 0.01) and *cPer3* (+52.94%, *p* < 0.01). In contrast, RL reduced the mesor of *cPer2* (−32.39%, *p* < 0.01) (Table 4; Figure 4D–G). GL significantly increased the overall expression of *cBmal1* and *cBmal2* when compared to RL (Figure 4H). Similarly, in addition to the *cCry1*, GL significantly increased the overall expression of negative clock gene, including *cCry2*, *cPer2*, and *cPer3* (Figure 4I).

### 3.5. Effect of Pinealectomy on the Circadian Rhythms of the Pancreas Clock Gene Expression in the Chick under Monochromatic Light

After pinealectomy, the expression of all clock genes retained daily variation (see Table 2 for *p* value) and circadian rhythm (see R^2^ and *p* value at Figure 5A–G). However, the expression and rhythmic parameters of clock genes changed significantly.

After pineal extraction, the total mRNA expression of all seven clock genes was significantly reduced under monochromatic light (Figure 5A–G). For rhythm parameters of clock genes, the acrophases of the *cClock* were advanced under monochromatic light compared with the control group (advance 1.23 h–3.50 h; *p* < 0.01), and *cBmal1* under BL advanced (−0.97 h; *p* < 0.01) and *cCry2* under GL advanced markedly (−3.56 h; *p* < 0.01) (Table 2), and the mesor values of all clock genes were significantly reduced under monochromatic light after pinealectomy (*cClock*: −12.71% to −43.43%; *cBmal1*: −33.33% to −39.53%; *cBmal2*: −4.5% to −18.66%; *cCry1*: −37.80% to −48.59%; *cCry2*: −35.71% to −66.32%; *cPer2*: −18.75% to −45.77%; *cPer3*: −20.51% to −39.81%, all *p* < 0.01), except that *cBmal2* under RL and BL changed very little (−4.50% to +5.62%) (Table 2, Figure 5). For amplitude, the amplitudes of all positive clock gene were decreased (Table 2, *cClock*: −25.97% to −33.33%; *cBmal1*: −38.89% to −50.0%; *cBmal2*: −29.27% to −50.0%, all *p* < 0.01), Similarly, the amplitudes of the negative clock gene were also greatly reduced (*cCry1*: −39.74% to −62.50%; *cCry2*: −40.83% to −72.34%; *cPer2*: −31.25% to −57.94%; *cPer3*: −21.05% to −48.08%, all *p* < 0.01) (Table 2). Overall, the expression of clock gene showed low−mesor and low−amplitude oscillations after pinealectomy, and the acrophase of the clock gene was slightly earlier. Monochromatic light had less effect on the mesor and amplitude of the clock gene after pinealectomy.

## 4. Discussion

### 4.1. Effects of Monochromatic Light on Carbohydrate Metabolism in Chicken Plasma

Acute exposure to bright light (>500–600 lux) in the evening increases insulin resistance and elevates postprandial insulin, glucose, and GLP-1 levels, relative to dim light (<2–5 lux) [29,30]. Another RCT reported that acute exposure to 2000-lux light in the evening impaired carbohydrate digestion [31]. One study proved that acute exposure to blue-enriched light (370 lux) in the evening has been found to increase peak postprandial glucose levels [32]. At present, no studies have shown the effect of monochromatic light on the rhythm of plasma insulin secretion. In this study, when chicks were exposed to monochromatic lights, RL significantly increased the mesor and amplitude of plasma insulin compared with GL. We found that plasma insulin was secreted rhythmically within 24 h; insulin levels peaked during the subjective day and then gradually decreased, reaching a minimum during the subjective night.

Our results revealed that plasma insulin levels were highest under RL and lowest under GL, but there were no significant differences in the insulin gene under different monochromatic light conditions. The insulin secretion is complex stimulus–secretion cascade. Priming and fusion are processes involved in the cascade of events that precedes Ca^2+^- dependent exocytosis and insulin secretion [33]. During the priming process, the soluble N-ethylmaleimide-sensitive factor attachment proteins (SNAREs) complex assembles to prepare the granules for Ca^2+^- dependent fusion. The plasma membrane-associated proteins syntaxin 1A (STX1A) and synaptosomal-protein of 25 kDa (SNAP25) interact with vesicle-associated membrane protein (VAMP2), which gives rise to a tertiary SNARE complex that promotes fusion by pulling the vesicle membrane in close contact with the plasma membrane upon Ca^2+^ influx [34]. We examined *cVamp2* mRNA expression and discovered that it was rhythmic and had the same tendency as insulin levels in plasma. The *cVamp2* mRNA expression was high in RL and low in GL, which was consistent with plasma insulin levels. This may suggest that changes of plasma insulin concentrations by monochromatic light depend on the release process rather than the synthesis pathway.

### 4.2. Effects of Melatonin on Monochromatic Light-Induced Insulin Concentration in Plasma

How do monochromatic lights affect the level of insulin in the blood plasma? Previous studies have shown that monochromatic light affects melatonin secretion, and green light increases plasma melatonin level [35]. However, it has been found that exposure to long-wavelength light for short periods of time at night does not inhibit the secretion of human melatonin [36]. This difference may be due to the length and timing of light exposure. Whether mice are able to produce endogenous melatonin is controversial and appears to be related to the strain of mice [37]. Jiang N. et al. have demonstrated that monochromatic light affects the expression of cAanat in the pineal gland of chicks through the biological clock, thereby affecting the synthesis of melatonin in the pineal gland [38]. Serotonin (5-HT) is a precursor to melatonin [39]. A study revealed that hens treated with RL had a higher concentration of 5-hydroxytryptamine (5-HT), a lower concentration of corticosterone (CORT) [40]. Wang et al. found that the number of 5HT-positive cells in the pineal gland of broiler chickens decreased under green light [41]. We hypothesized that red light reduces the conversion of serotonin to melatonin by reducing the expression of Aanat, leading to the accumulation of serotonin.

Reports on the interactions between melatonin and the glucose metabolism of rats and humans have shown phenomenological and functional causal-analytic results. In a relatively early stage of type 2 diabetes, insulin secretion is increased and melatonin is decreased—a pattern that is observed in rats and humans. The mRNA transcript levels of melatonin receptors appear to be significantly higher in type 2 diabetic patients than in control groups [42]. The insulin receptor mRNA of the pineal gland was found to be reduced in type 2 diabetic rats, suggesting a functional relationship between melatonin and insulin [43]. Melatonin release and metabolism are both impacted by blindness and even reduced light perception caused by the loss of retinal ganglion cells [44,45]. However, there is a disagreement on whether melatonin may be metabolically beneficial or deleterious, one study noted that the effect of melatonin on glucose tolerance seems to depend on the timing of the diet, therefore, the relative timing between elevated melatonin concentrations and glycemic challenges should be considered [46]. Melatonin regulates energy metabolism, pancreatic function, and remodeling in rats during pregnancy and early lactation, and a lack of melatonin during pregnancy can impair glucose metabolism, demonstrating melatonin’s effect on regulating insulin secretion [47]. However, the effect of melatonin on the circadian secretion of mammalian insulin depends mainly on MT1 and MT2 membrane receptors [48]. MT1- and/or MT2-mediated melatonin action decreases glucose-stimulated insulin secretion in isolated rat pancreatic islets and rat insulinoma beta-cells [49]. The study found that melatonin membrane receptors had different locations in mouse islets, MT1 is expressed in α-cells while MT2 is located to the β-cells [50]. There are three melatonin membrane receptors in chickens, called Mel1a, Mel1b, and Mel1c [26]. Unfortunately, our work did not reveal the distribution of melatonin receptors in chicken islets. Furthermore, it is known that MT2 receptor mutations are linked to elevated fasting glucose [51] as well as prolonged melatonin secretion [52], which may be affected by light perception [44]. Presently, the effect of melatonin on insulin in poultry has not been reported. In this study, RL significantly reduced the mesor and amplitude of plasma melatonin levels compared to GL. Under monochromatic light, plasma melatonin levels showed a rhythmic expression of day low and night high and was significantly negatively correlated with the daily variation of plasma insulin. In order to verify that melatonin may play a mediating role in the effect of monochromatic light on plasma insulin, we established a melatonin deficiency model by performing pineal removal surgery. We found that pineal removal resulted in low-mesor and low-amplitude oscillation of the plasma melatonin levels. In contrast, after pinealectomy, plasma insulin levels increased significantly.

### 4.3. Role of Clock Gene in Melatonin-Mediated Monochromatic Light-Induced Insulin Rhythm in Plasma

As melatonin is secreted in a clearly diurnal fashion, it is safe to assume that it also has a diurnal impact on the blood-glucose-regulating function of the islet [53]. Circadian expression of clock genes (Clock, Bmal1, Per1,2,3, and Cry1,2) in pancreatic islets and INS1 rat insulinoma cells may indicate that circadian rhythms are generated within the b-cells. The circadian secretion of insulin from pancreatic islets is clock-driven. Circadian rhythm and clock function disruption leads to metabolic disturbances, such as type 2 diabetes. However, the clock genes in the poultry pancreatic tissue have not been reported. Previous studies have shown that monochromatic lights can affect the expression of clock genes in the poultry central oscillator hypothalamus [24], retina [23], and peripheral oscillator liver [25]. In this study, we found that monochromatic light affected the transcription of the clock gene in broiler pancreas, GL increased the mesor values of the positive clock genes (*cClock*, *cBmal1* and *cBmal2*) and the negative clock genes (*cCry2*, *cPer3*) (Table 4). Moreover, compared to RL, GL significantly increased the expression of clock gene (*cBmal1*, *cBmal2*, *cCry2*, *cPer2* and *cPer3*) (Figure 6H-I) except for the clock genes *cCry1* and *cCry2*, the mesor and amplitude of insulin level showed a significant negative correlation with the clock genes. The circadian secretion of insulin concentration from pancreatic islets is clock-driven. Disruption of circadian rhythms and clock function leads to metabolic disturbances, for example, mice that specifically knocked out *Bmal1* in the pancreas exhibited blood glucose abnormalities, as well as defects in insulin response after glucose stimulation. In addition, overall reduced insulin secretion from isolated islets was monitored, although the insulin content in BMAL1-positive and BMAL1-negative islets did not differ [16]. In Clock^19/19^ mutant mice, elevated glucose levels across the entire circadian period were also displayed, with an inhibited insulin response at the beginning of the feeding period [54]. A few studies have shown abnormal expression of the islet core clock gene in patients with type 2 diabetes. Stamenkovic et al. [55] reported that *Per2*, *Per3* and *Cry2* were downregulated in islets from type 2 diabetic donors compared to islets of normoglycemic controls.

In conclusion, our results suggested that RL decreased the rhythm of plasma melatonin level and transcription of the pancreatic clock gene, stimulating insulin secretion by increasing the expression of *cVamp2* but not *cInsulin* (Figure 6). Thus, we speculate that melatonin may play a key role in the effect of monochromatic light on the rhythm of plasma insulin concentration by affecting the expression of the clock genes in the chicken pancreas.

## Figures and Tables

**Figure 1 ijms-24-02368-f001:**
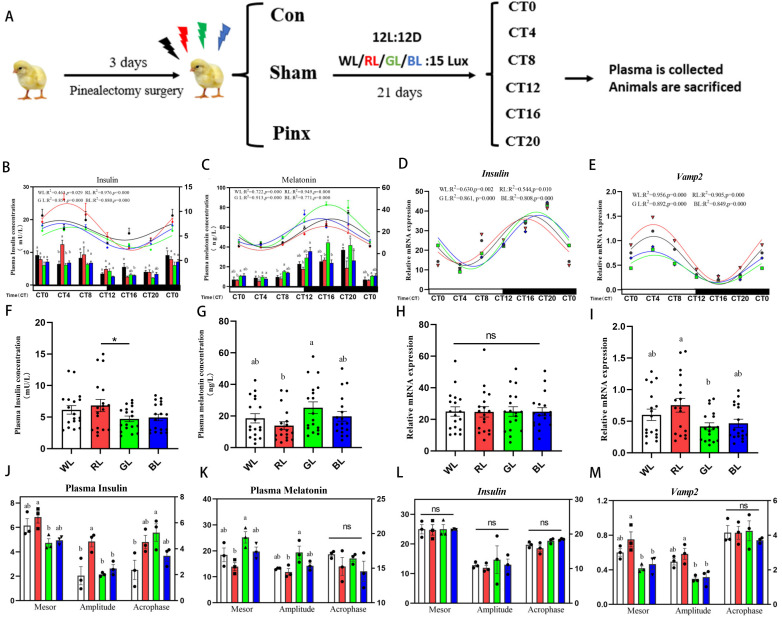
Circadian rhythms of plasma insulin, melatonin and pancreatic insulin-related gene in the chicks under monochromatic light. (**A**): Schematic diagram of animal experiments. Temporal changes of plasma insulin (**B**), plasma melatonin (**C**), pancreatic *cInsulin* (**D**) and *cVamp2* (**E**). The column represents the insulin concentration level (uM/L) (left Y-axis) and the line with symbols represents the 24-h circadian rhythm (right Y-axis). plasma insulin (**F**) and melatonin (**G**) concentration, mRNA expression of *cInsulin* (**H**) and *cVamp2* (**I**), rhythmic parameters of plasma insulin (**J**), melatonin (**K**), *cInsulin* (**L**) and *cVamp2* (**M**) (the mesor and amplitude according to left Y-axis, the acrophase according to right Y-axis). The horizontal white bar on each figure represents the subjective day, and the black bar represents the subjective night. Quantitative analysis of the PCR data is shown as the mean ± SEM. * *p* < 0.05, ns: no significance. CT, circadian time. CON: intact group; Sham: sham operation group; Pinx: pinealectomy group. The curve indicates the best fit to the points by cosinor analysis. R^2^ values represent the degree of fitting, *p*-values indicate the significance of regression analysis, with significance defined as *p* < 0.05. Different letters indicate significant differences among the monochromatic light groups, and the light treatments are represented with different colors (black, WL; red, RL; green, GL; blue, BL).

**Figure 2 ijms-24-02368-f002:**
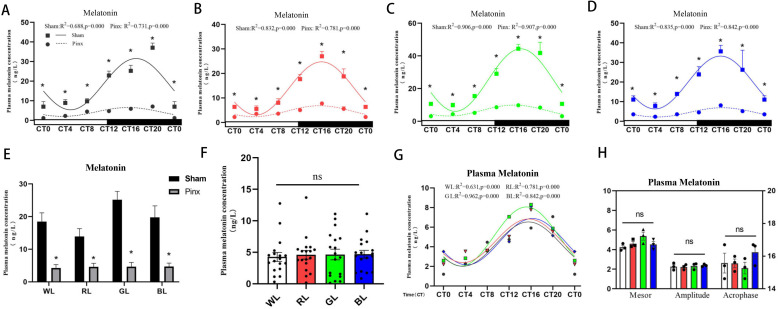
Circadian rhythms of plasma melatonin in chicks after pinealectomy. The solid line with square symbols represents the circadian rhythm in the sham operation group, and the dashed line with circle symbols represents the circadian rhythm in the pinealectomy group. Goodness of rhythmicity (R^2^ and *p* value) is shown at the top of each figure. (**A**): WL, (**B**): RL, (**C**): GL, (**D**): BL, (**E**): changes of plasma melatonin level, (**D**): plasma melatonin under monochromatic light in the pinealectomy group, (**E**): rhythmic expression of plasma melatonin was merged together after pineal removal, (**H**): rhythm parameters of melatonin (the mesor and amplitude according to left Y-axis, the acrophase according to right Y-axis). The horizontal white bar on each figure represents the subjective day, and the black bar represents the subjective night. Quantitative analysis of the PCR data is shown as the mean ± SEM. CT, circadian time. The asterisk “*” indicates significant differences between the sham operation group and the pinealectomy group. ns: no significance. The light treatments are represented with different colors (black, WL; red, RL; green, GL; blue, BL).

**Figure 3 ijms-24-02368-f003:**
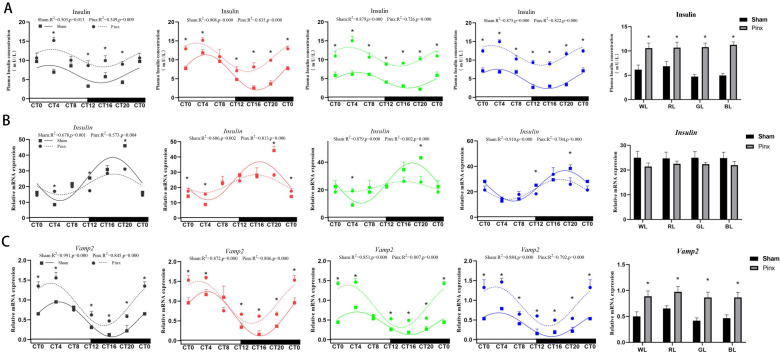
Rhythmic expression of plasma insulin and insulin-related gene in the chick pancreas after pinealectomy. The dashed line with circle symbols represents the circadian rhythm in the pinealectomy group, and the solid line with square symbols represents the circadian rhythm in the sham operation group. Goodness of rhythmicity (R^2^ and *p* value) is shown at the top of each figure. (**A**): plasma insulin, (**B**): *cInsulin*, (**C**): *cVamp2*. *Vamp2*: vesicle-associated membrane protein 2. The horizontal white bar on each figure represents the subjective day, and the black bar represents the subjective night. The column represents the total plasma insulin at the six time points. Quantitative analysis of the PCR data is shown as the mean ± SEM. CT, circadian time. The asterisk “*” indicates significant differences between the sham operation group and the pinealectomy group. The light treatments are represented with different colors (black, WL; red, RL; green, GL; blue, BL).

**Figure 4 ijms-24-02368-f004:**
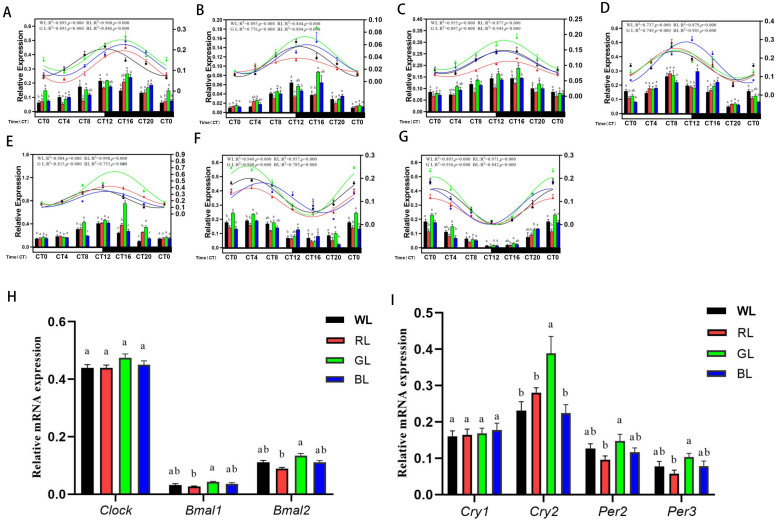
Circadian rhythms of pancreas clock gene transcription in the chicks under monochromatic light (12 L: 12 D). (**A**), *cClock*; (**B**), *cBmal1*; (**C**), *cBmal2*; (**D**), *cCry1*; (**E**), *cCry2*; (**F**), *cPer2*; (**G**), *cPer3*; the column represents clock gene expression level (left Y-axis), and the line with symbols represents the 24-h circadian rhythm (right Y-axis). The goodness of rhythmicity (R^2^ and *p* value) is shown on the top of each figure. (**H**): the total mRNA expression of the positive clock gene; (**I**): the total mRNA expression of the negative clock gene. The horizontal white bar on each figure represents the subjective day, and the black bar represents the subjective night. Quantitative analysis of the PCR data is shown as the mean ± SEM. CT, circadian time. Different letters indicate significant differences among the monochromatic light groups, and the light treatments are represented with different colors (black, WL; red, RL; green, GL; blue, BL).

**Figure 5 ijms-24-02368-f005:**
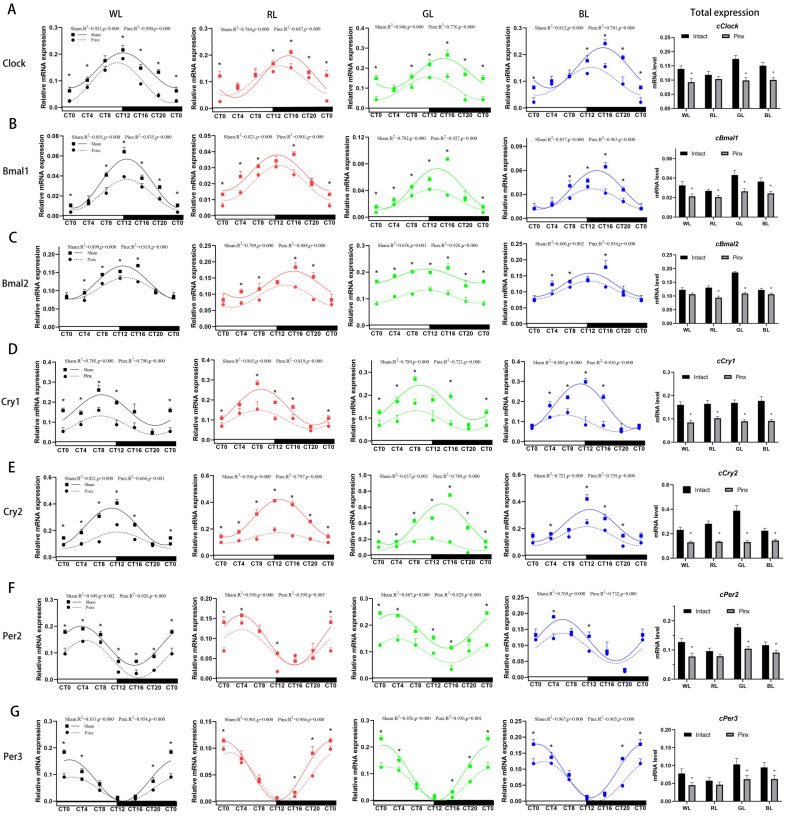
Circadian rhythms of pancreas clock gene transcription in chicks after pinealectomy. The dashed line with circle symbols represents the circadian rhythm in the pinealectomy group, and the solid line with square symbols represents the circadian rhythm in the sham operation group. Goodness of rhythmicity (R^2^ and *p* value) is shown at the top of each figure. The column represents the total mRNA expression at the six time points. (**A**), *cClock*; (**B**), *cBmal1*; (**C**), *cBmal2*; (**D**), *cCry1*; (**E**), *cCry2*; (**F**), *cPer2*; (**G**), *cPer3*. The horizontal white bar on each figure represents the subjective day, and the black bar represents the subjective night. Quantitative analysis of the PCR data is shown as the mean ± SEM. CT, circadian time. The asterisk “*” indicates significant differences between the sham operation group and the pinealectomy group. The light treatments are represented with different colors (black, WL; red, RL; green, GL; blue, BL).

**Figure 6 ijms-24-02368-f006:**
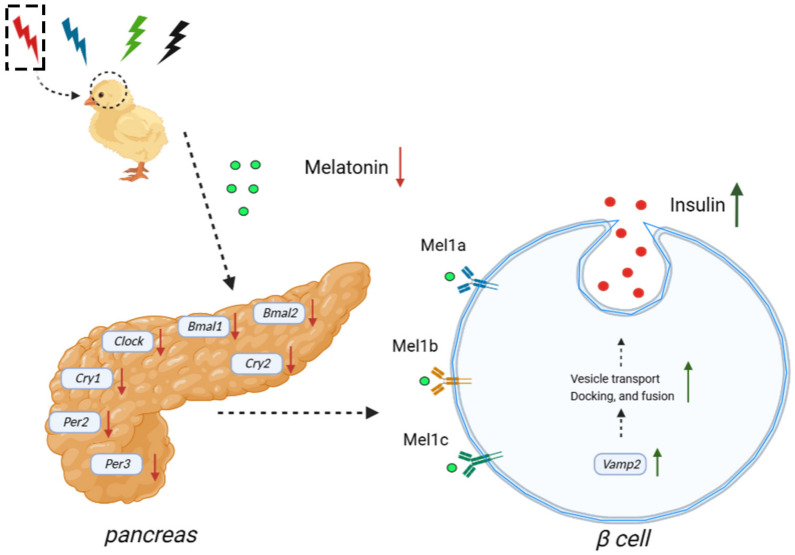
A schematic of our proposed model. Effects of monochromatic light on pancreatic clock genes and rhythmic secretion of insulin in chickens. RL decreased the rhythm of plasma melatonin level and transcription of pancreatic clock gene, stimulating insulin secretion by increasing the expression of *cVamp2*.

**Table 1 ijms-24-02368-t001:** Primers used for real-time PCR analysis and expected product length.

Gene	Accession No.	Primer Sequence (5′ to 3′)		Length (bp)
*cClock*	NM_204174.2	F: GATCACAGGGCACCTCCAATA	R: CTAGTTCTCGCCGCCTTTCT	301
*cBmal1*	NM_001001463.1	F: GTAGACCAGAGGGCGACAG	R: ATGAAACTGAACCAGCGACTC	215
*cBmal2*	NM_204133.1	F: CGGCGTTCCTTCTTCTGTC	R: TTCCTCTTCCACTCCCACC	165
*cCry1*	NM_204245.1	F: GATGTGGCTATCCTGTAGTTCCT	R: GCTGCTGGTAGATTTGTTTCAT	281
*cCry2*	NM_204244.1	F: GCACGGCTGGATAAACACT	R: AAATAAGCGGCAGGACAAA	141
*cPer2*	NM_204262.1	F: ATGAAACGAGCCATCCCG	R: CAGTTGTCGTGATTTTGCCTA	206
*cPer3*	XM_417528.2	F: CAGTGCCTTTGTTGGGTTAC	R: GATGGATTCACAAAACTGGAC	217
*cInsulin*	NM_205222.3	F: TCTTTTCTGGCCCTGGAACC	R: GCAAGGGACTGCTCACTAGG	159
*cVamp2*	XM_040679081.2	F: AAAATGTGCACGCACTGACC	R: CCTCGTGTCTGCTGAAACCT	153
*cGAPDH*	NM_204305.1	F: ATCACAGCCACACAGAAGACG	R: TGACTTTCCCCACAGCCTTA	124

*Clock*: Circadian locomotor output cycles kaput; *Bmal1/2*: Brain and muscle Arnt-like 1/2; *Cry1/2*: Cryptochrome 1/2; *Per2/3*: Period 2/3; *Vamp2*: vesicle—associated membrane protein 2; F = forward primer; R = reverse primer.

**Table 2 ijms-24-02368-t002:** The daily temporal difference parameters and circadian rhythm parameters of plasma melatonin and pancreas clock gene transcription under different monochromatic lights after pinealectomy.

		MT	*cClock*	*cBmal1*	*cBmal2*	*cCry1*	*cCry2*	*cPer2*	*cPer3*
Mesor	WPx	4.245 ± 0.163 ^a^*	0.093 ± 0.001 ^a^*	0.021 ± 0.002 ^a^*	0.106 ± 0.002 ^a^*	0.085 ± 0.003 ^a^*	0.129 ± 0.006 ^a^*	0.077 ± 0.004 ^b^*	0.045 ± 0.008 ^a^*
	RPx	4.598 ± 0.190 ^a^*	0.103 ± 0.002 ^a^*	0.021 ± 0.005 ^a^*	0.094 ± 0.006 ^a^	0.102 ± 0.002 ^b^*	0.134 ± 0.005 ^a^*	0.078 ± 0.005 ^b^*	0.046 ± 0.003 ^a^*
	GPx	5.377 ± 0.338 ^a^*	0.099 ± 0.001 ^a^*	0.026 ± 0.001 ^a^*	0.109 ± 0.004 ^a^*	0.089 ± 0.006 ^ab^*	0.131 ± 0.003 ^a^*	0.104 ± 0.009 ^a^*	0.062 ± 0.004 ^a^*
	BPx	4.527 ± 0.242 ^a^*	0.100 ± 0.004 ^a^*	0.024 ± 0.002 ^a^*	0.106 ± 0.001 ^a^	0.091 ± 0.002 ^ab^*	0.144 ± 0.002 ^a^*	0.091 ± 0.002 ^ab^*	0.062 ± 0.003 ^a^*
Amplitude	WPx	2.267 ± 0.169 ^a^*	0.075 ± 0.013 ^a^	0.015 ± 0.002 ^a^*	0.029 ± 0.003 ^a^*	0.047 ± 0.012 ^a^*	0.061 ± 0.009 ^a^*	0.050 ± 0.007 ^a^*	0.046 ± 0.008 ^a^*
	RPx	2.225 ± 0.125 ^a^*	0.055 ± 0.005 ^a^*	0.011 ± 0.003 ^a^*	0.034 ± 0.006 ^a^*	0.046 ± 0.005 ^a^*	0.039 ± 0.014 ^a^*	0.044 ± 0.003 ^a^*	0.045 ± 0.005 ^a^*
	GPx	2.330 ± 0.159 ^a^*	0.057 ± 0.002 ^a^*	0.015 ± 0.004 ^a^*	0.024 ± 0.005 ^a^*	0.043 ± 0.002 ^a^*	0.073 ± 0.004 ^a^*	0.045 ± 0.006 ^a^*	0.054 ± 0.008 ^a^*
	BPx	2.386 ± 0.059 ^a^*	0.052 ± 0.003 ^a^*	0.013 ± 0.002 ^a^*	0.028 ± 0.009 ^a^*	0.042 ± 0.004 ^a^*	0.071 ± 0.008 ^a^*	0.047 ± 0.008 ^a^*	0.051 ± 0.003 ^a^*
Acrophase	WPx	15.58 ± 0.496 ^a^	10.77 ± 1.465 ^a^*	12.57 ± 0.346 ^a^	13.21 ± 0.120 ^a^	8.437 ± 0.563 ^a^	12.10 ± 0.245 ^ab^	4.570 ± 0.130 ^a^	24.818 ± 0.123 ^a^
(CT)	RPx	15.56 ± 0.219 ^a^	13.44 ± 0.386 ^a^*	12.72 ± 0.745 ^a^*	13.74 ± 0.652 ^a^	8.269 ± 0.488 ^a^	13.02 ± 0.132 ^a^	3.947 ± 0.107 ^a^	24.439 ± 0.687 ^a^
	GPx	15.27 ± 0.292 ^a^	11.96 ± 0.265 ^a^*	12.59 ± 0.260 ^a^	11.23 ± 0.874 ^a^	8.276 ± 0.360 ^a^	10.74 ± 0.211 ^b^*	3.823 ± 0.854 ^a^	24.385 ± 0.798 ^a^
	BPx	16.24 ± 0.340 ^a^	12.80 ± 0.544 ^a^*	12.96 ± 0.380 ^a^*	12.30 ± 0.477 ^a^	7.354 ± 0.142 ^a^*	11.88 ± 0.132 ^ab^	5.944 ± 0.963 ^a^	25.301 ± 0.880 ^a^
*p*	WPx	0	0	0	0	0	0.001	0	0
	RPx	0	0	0	0	0	0	0.004	0
	GPx	0	0	0	0	0	0	0	0
	BPx	0	0	0	0	0	0	0	0

Daily temporal differences were analyzed by SPSS 20 with one-way ANOVA. Rhythm parameters (mesor, amplitude, acrophase) were generated by Originlab Software 9.3, and were shown as the Mean ± SEM. MT: melatonin; Different letters indicate significant differences among the monochromatic light groups and “*” indicate significant differences between pinealectomy group and control group.

**Table 3 ijms-24-02368-t003:** The daily temporal difference parameters and circadian rhythm parameters of pancreas *cVamp2* and *cInsulin* transcription under different monochromatic lights in the control and pinealectomy group.

		*cVamp2*	*cInsulin*		*cVamp2*	*cInsulin*
Mesor	WL	0.601 ± 0.036 ^ab^	24.95 ± 1.281 ^a^	WPx	0.859 ± 0.021 *^a^	21.41 ± 0.429 ^a^
	RL	0.752 ± 0.071 ^a^	24.72 ± 1.440 ^a^	RPx	0.943 ± 0.019 *^a^	22.65 ± 0.307 ^a^
	GL	0.418 ± 0.022 ^b^	24.95 ± 1.290 ^a^	GPx	0.866 ± 0.025 *^a^	22.28 ± 1.023 ^a^
	BL	0.465 ± 0.051 ^b^	24.82 ± 0.101 ^a^	BPx	0.867 ± 0.050 *^a^	21.91 ± 2.048 ^a^
Amplitude	WL	0.494 ± 0.031 ^ab^	12.83 ± 0.488 ^a^	WPx	0.546 ± 0.012 ^a^	6.624 ± 1.355 *^a^
	RL	0.584 ± 0.053 ^a^	11.91 ± 0.488 ^a^	RPx	0.541 ± 0.066 ^a^	7.204 ± 0.823 *^a^
	GL	0.294 ± 0.018 ^b^	14.66 ± 0.894 ^a^	GPx	0.541 ± 0.027 *^a^	6.851 ± 0.757 *^a^
	BL	0.314 ± 0.045 ^b^	13.00 ± 1.545 ^a^	BPx	0.499 ± 0.081 ^a^	8.403 ± 3.476 ^a^
Acrophase	WL	4.153 ± 0.335 ^a^	16.89 ± 0.396 ^a^	WPx	2.818 ± 0.426 ^a^	17.42 ± 0.025 ^a^
(CT)	RL	4.141 ± 0.303 ^a^	15.90 ± 0.757 ^a^	RPx	2.496 ± 0.413 *^a^	16.07 ± 1.418 ^a^
	GL	4.254 ± 0.472 ^a^	18.05 ± 0.360 ^a^	GPx	2.222 ± 0.184 *^a^	18.07 ± 0.650 ^a^
	BL	3.711 ± 0.079 ^a^	18.45 ± 0.120 ^a^	BPx	2.680 ± 0.276 *^a^	16.45 ± 0.899 ^a^
*p*	WL	0	0.002	WPx	0	0.006
	RL	0	0.01	RPx	0	0
	GL	0	0	GPx	0	0
	BL	0	0	BPx	0	0

Daily temporal differences were analyzed by SPSS 20 with one-way ANOVA. Rhythm parameters (mesor, amplitude, acrophase) were generated by Originlab Software 9.3, and were shown as the Mean ± SEM. *Vamp2*: vesicle-associated membrane protein 2. Different letters indicate significant differences among the monochromatic light groups and “*” indicate significant differences between pinealectomy group and control group.

**Table 4 ijms-24-02368-t004:** The daily temporal difference parameters and circadian rhythm parameters of plasma melatonin and pancreas clock gene transcription under different monochromatic lights.

		MT	*cClock*	*cBmal1*	*cBmal2*	*cCry1*	*cCry2*	*cPer2*	*cPer3*
Mesor	WL	18.47 ± 2.155 ^ab^	0.139 ± 0.004 ^bc^	0.032 ± 0.002 ^b^	0.111 ± 0.006 ^b^	0.160 ± 0.011 ^a^	0.266 ± 0.007 ^b^	0.142 ± 0.005 ^ab^	0.068 ± 0.008 ^b^
	RL	13.90 ± 1.266 ^b^	0.118 ± 0.003 ^c^	0.032 ± 0.001 ^b^	0.089 ± 0.002 ^c^	0.164 ± 0.002 ^a^	0.281 ± 0.007 ^b^	0.096 ± 0.009 ^c^	0.058 ± 0.003 ^b^
	GL	25.19 ± 1.786 ^a^	0.175 ± 0.006 ^a^	0.043 ± 0.003 ^a^	0.134 ± 0.004 ^a^	0.168 ± 0.005 ^a^	0.389 ± 0.006 ^a^	0.148 ± 0.006 ^a^	0.103 ± 0.005 ^a^
	BL	19.77 ± 1.400 ^ab^	0.151 ± 0.006 ^b^	0.036 ± 0.003 ^ab^	0.111 ± 0.002 ^b^	0.177 ± 0.002 ^a^	0.224 ± 0.006 ^c^	0.116 ± 0.007 ^b^	0.078 ± 0.002 ^b^
Amplitude	WL	13.18 ± 0.166 ^ab^	0.073 ± 0.009 ^ab^	0.025 ± 0.002 ^a^	0.041 ± 0.003 ^ab^	0.078 ± 0.005 ^a^	0.123 ± 0.015 ^b^	0.102 ± 0.003 ^ab^	0.068 ± 0.005 ^bc^
	RL	11.86 ± 1.104 ^b^	0.069 ± 0.003 ^b^	0.018 ± 0.001 ^b^	0.026 ± 0.004 ^b^	0.091 ± 0.007 ^a^	0.141 ± 0.006 ^b^	0.064 ± 0.009 ^b^	0.057 ± 0.005 ^c^
	GL	19.38 ± 2.020 ^a^	0.077 ± 0.002 ^a^	0.030 ± 0.002 ^a^	0.048 ± 0.006 ^a^	0.077 ± 0.014 ^a^	0.254 ± 0.012 ^a^	0.107 ± 0.004 ^a^	0.104 ± 0.003 ^a^
	BL	14.26 ± 0.826 ^ab^	0.076 ± 0.009 ^ab^	0.024 ± 0.002 ^ab^	0.035 ± 0.003 ^ab^	0.112 ± 0.004 ^a^	0.120 ± 0.018 ^b^	0.080 ± 0.017 ^ab^	0.077 ± 0.005 ^b^
Acrophase	WL	16.99 ± 0.244 ^a^	12.0 ± 0.223 ^a^	12.56 ± 0.261 ^b^	13.36 ± 0.867 ^a^	8.615 ± 0.271 ^b^	10.82 ± 0.499 ^a^	4.243 ± 0.137 ^b^	24.42 ± 0.586 ^a^
(CT)	RL	15.25 ± 1.053 ^a^	15.5 ± 0.272 ^b^	10.79 ± 0.231 ^a^	14.51 ± 0.678 ^a^	8.780 ± 0.360 ^b^	13.22 ± 0.249 ^bc^	3.883 ± 0.454 ^b^	24.13 ± 0.405 ^a^
	GL	16.41 ± 0.352 ^a^	15.4 ± 0.595 ^b^	13.68 ± 0.335 ^b^	13.63 ± 0.332 ^a^	9.489 ± 0.508 ^ab^	14.30 ± 0.117 ^b^	3.068 ± 0.185 ^b^	24.55 ± 0.259 ^a^
	BL	14.54 ± 1.196 ^a^	15.5 ± 0.467 ^b^	13.93 ± 0.550 ^b^	13.15 ± 0.347 ^a^	10.79 ± 0.259 ^a^	12.52 ± 0.370 ^c^	7.175 ± 1.297 ^a^	23.46 ± 0.258 ^a^
*p*	WL	0	0	0	0	0	0	0	0
	RL	0	0	0	0	0	0	0	0
	GL	0	0	0	0	0	0	0	0
	BL	0	0	0	0	0	0	0	0

Daily temporal differences were analyzed by SPSS 20 with one-way ANOVA. Rhythm parameters (mesor, amplitude, acrophase) were generated by Originlab Software 9.3, and were shown as the mean ± SEM. MT: melatonin; Different letters indicate significant differences among the monochromatic light groups.

## Data Availability

Data are contained within the article.

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
