# Peer review of "Role of Melatonin in Daily Variations of Plasma Insulin Level and Pancreatic Clock Gene Expression in Chick Exposed to Monochromatic Light"

_ijms, 2023, doi:10.3390/ijms24032368_

Round 1
Reviewer 1 Report
In this interesting paper Chen et al. demonstrate how red, green and blue light exposure at 12:12 L:D alternation affect melatonin, insulin, cVamp2 and clock genes expression in control and after pinealectomy. Results allowed authors to assume that melatonin may play a key role in the effect of monochromatic light on the circadian rhythm of plasma insulin via the expression of clock gene in chicken pancreas.
I can recommend this papers for publication, however the following issues should be considered first:
MAJOR
1. The authors are invited to discuss in greater details the putative mechanisms behind different effects of red vs blue/green light and its putative differences between birds and mammals (chicks and humans). For example, in humans it was shown that red light has no effect on melatonin secretion (doi: 10.1081/cbi-100107515). What about possibility that red light exposure may cause lower melatonin mesor and amplitude via preventing production of its precursor, serotonin which is blue light-dependent (e.g.,https://www.ncbi.nlm.nih.gov/pmc/articles/PMC6746555/)?
2. Is there a difference in melanopic sensitivity in birds (chicks) vs mammalian species that are most sensitive to blue rather than green light (peak spectral sensitivity, 480 nm, DOI: 10.1177/0748730411409719)?
3. Also In this context citation of work by Cippola Neto & Amaral (doi.org/10.1210/er.2018-00084) to support the sentence “Previous studies have shown that monochromatic light affects melatonin secretion, and green light increases plasma melatonin level” (Discussion, P.12, L.369-371) is not justified. The authors refer not to green, but to blue light effects which de facto decrease, not increase nocturnal melatonin. Putative increase melatonin, as mentioned above, is possible via increase in daytime synthesis of its light-dependent precursors. This should be dicussed in details.
4. Furthermore, in this context: in the discussion authors wrote: “… high melatonin levels, due to blindness [45] or through exogenous application of melatonin, raise blood glucose levels [46], whereas a decrease in blood glucose and an increase in insulin levels are observed after pinealectomy [47]”. The cited papers are from the 70s and another crucial interelated points should be considered: a) relative timing between elevated melatonin concentrations and glycemic load, doi: 10.1016/j.tem.2019.11.011, since there are well known phase response curves to light, melatonin that are also present for metabolic endpoints (doi: 10.1038/s41467-022-28308-6); b) not mere amount of perceived light, or light intensity, but rather its amplitude between daytime and night is the most important facrot for maintenance of melatonin production and its output on circadian metabolic functions. Blindness and even diminshed light perception due to retinal ganglion cells loss affects both melatonin secretion (doi: 10.1111/jpi.12730; doi:10.4103/1673-5374.332149) and metabolism (https://doi.org/10.3390/app12136374) that can be mitigated by timed melatonin (doi: 10.1111/jpi.12730; doi:10.4103/1673-5374.332149).
5. Obviously, average plasma melatonin levels in chicks are much higher than in the mammalian species and in humans (lower than 5 pg/ml during daytime and about 10-fold higher by night, e.g., https://doi.org/10.1111/jpi.12657). Information on what values for melatonin parameters (MESOR, amplitude phase) should be considered the reference values would be helpful to better comprehend the results of this study. Also, how much these values depend on age or gender?
6. The authors speculate that melatonin may mediate effects of monochromatic light on the circadian rhythm of plasma insulin via the expression of clock gene in chicken pancreas. However, melatonin effect on circadian insulin production at least in mammals including humans is mainly attributed to MT1 and MT2 membrane receptors (see e.g., already refenced work#49, see as well doi: 10.1590/2359-3997000000098). Furthermore, it is known that mutation in MT2 receptor is linked to elevated fasting glucose (doi:10.1038/ng.290) along with prolonged melatonin secretion (doi: 10.2337/db15-0999) that may depend on the light perception (doi: 10.1111/jpi.12730). The authors do not at all mention melatonin receptors in Discussion, though, I believe, this issue deserves detailed discussion.
MINOR
7. Material and methods. Providing information on light intensity (also) in lux rather than in W/m2 might be easier for readers to comprehend.
8. Quality of figures should be increased as not all text with the statistical details is readable.
Author Response
Dear Editor and Reviewers,
I’m very glad to hear from you, and thank you for your work in dealing with my manuscript. All of those comments are valuable and very helpful for improving the quality of our manuscript and the important guidance to our research. We have carefully checked the manuscript and revised it according to the suggestions. Now, I’m submitting here a reply to queries.
Reviewer #1:
In this interesting paper Chen et al. demonstrate how red, green and blue light exposure at 12:12 L:D alternation affect melatonin, insulin, cVamp2 and clock genes expression in control and after pinealectomy. Results allowed authors to assume that melatonin may play a key role in the effect of monochromatic light on the circadian rhythm of plasma insulin via the expression of clock gene in chicken pancreas.
I can recommend this papers for publication, however the following issues should be considered first:
MAJOR
Q1. The authors are invited to discuss in greater details the putative mechanisms behind different effects of red vs blue/green light and its putative differences between birds and mammals (chicks and humans). For example, in humans it was shown that red light has no effect on melatonin secretion (doi: 10.1081/cbi-100107515). What about possibility that red light exposure may cause lower melatonin mesor and amplitude via preventing production of its precursor, serotonin which is blue light-dependent (e.g.,https://www.ncbi.nlm.nih.gov/pmc/articles/PMC6746555/)?
Answer: Thank you for your suggestion. We have added the effects of different wavelengths of light on serotonin and its putative differences between birds and mammals in the discussion part. Please see Page 16 Line 433-437 and Line 440-445 in the revised manuscript R1.
Q2. Is there a difference in melanopic sensitivity in birds (chicks) vs mammalian species that are most sensitive to blue rather than green light (peak spectral sensitivity, 480 nm, DOI: 10.1177/0748730411409719)?
Answer: Thank you for your question. Intrinsically photosensitive ganglion cells (ipRGCs) in retina contained melanopsin, which entrains to environmental light cycles for mammal [1]. Nonmammalian vertebrates include two melanopsin genes, Opn4-1 (exclusively expressed in nonmammalian vertebrates, also called Opn4x) and Opn4-2 (present in mammals and all other vertebrates, also named Opn4m) [2]. One study showed that green light promoted the circadian expression of cOpn4-1, cOpn4-2 as well as the secretion of retinal melatonin, with increased their mesors and amplitudes [3].
References:
- Hankins MW, Peirson SN, Foster RG. Melanopsin: an exciting photopigment. Trends Neurosci 2008, 31(1), 27-36, doi: 10.1016/j.tins.2007.11.002.
- Bellingham J, Chaurasia SS, Melyan Z, Liu C, Cameron MA, Tarttelin EE, Iuvone PM, Hankins MW, Tosini G, Lucas RJ. Evolution of melanopsin photoreceptors: discovery and characterization of a new melanopsin in nonmammalian vertebrates. PLoS Biol 2006, 4(8), e254, doi: 10.1371/journal.pbio.0040254.
- Bian J, Wang Z, Dong Y, Cao J, Chen Y. Effect of monochromatic light on the circadian clock of cultured chick retinal tissue. Exp Eye Res 2020, 194, 108008, doi: 10.1016/j.exer.2020.108008.
Q3. Also In this context citation of work by Cippola Neto & Amaral (doi.org/10.1210/er.2018-00084) to support the sentence “Previous studies have shown that monochromatic light affects melatonin secretion, and green light increases plasma melatonin level” (Discussion, P.12, L.369-371) is not justified. The authors refer not to green, but to blue light effects which de facto decrease, not increase nocturnal melatonin. Putative increase melatonin, as mentioned above, is possible via increase in daytime synthesis of its light-dependent precursors. This should be dicussed in details.
Answer: Thank you for your suggestion. We have removed references “Melatonin as a Hormone: New Physiological and Clinical Insights”. In the discussion part, we have illustrated the effects of different light wavelengths on serotonin, which rises in response to red light [1] and decreases in response to green light [2]. We hypothesized that red light reduces the conversion of serotonin to melatonin by reducing the expression of Aanat [3], leading to the accumulation of serotonin. Please see Page 16 Line 440-445 in the revised manuscript R1.
References:
- Shi, H.; Li, B.; Tong, Q.; Zheng, W.; Zeng, D.; Feng, G. Effects of LED Light Color and Intensity on Feather Pecking and Fear Responses of Layer Breeders in Natural Mating Colony Cages. Animals (Basel) 2019, 9, doi:10.3390/ani9100814.
- Jin, E.; Jia, L.; Li, J.; Yang, G.; Wang, Z.; Cao, J.; Chen, Y. Effect of monochromatic light on melatonin secretion and arylalkylamine N-acetyltransferase mRNA expression in the retina and pineal gland of broilers. Anat Rec (Hoboken) 2011, 294, 1233-1241, doi:10.1002/ar.21408.
- Jiang, N.; Cao, J.; Wang, Z.; Dong, Y.; Chen, Y. Effect of monochromatic light on the temporal expression of N-acetyltransferase in chick pineal gland. Chronobiol Int 2020, 37, 1140-1150, doi:10.1080/07420528.2020.1754846.
Q4. Furthermore, in this context: in the discussion authors wrote: “… high melatonin levels, due to blindness [45] or through exogenous application of melatonin, raise blood glucose levels [46], whereas a decrease in blood glucose and an increase in insulin levels are observed after pinealectomy [47]”. The cited papers are from the 70s and another crucial interelated points should be considered: a) relative timing between elevated melatonin concentrations and glycemic load, doi: 10.1016/j.tem.2019.11.011, since there are well known phase response curves to light, melatonin that are also present for metabolic endpoints (doi: 10.1038/s41467-022-28308-6); b) not mere amount of perceived light, or light intensity, but rather its amplitude between daytime and night is the most important facrot for maintenance of melatonin production and its output on circadian metabolic functions. Blindness and even diminshed light perception due to retinal ganglion cells loss affects both melatonin secretion (doi: 10.1111/jpi.12730; doi:10.4103/1673-5374.332149) and metabolism (https://doi.org/10.3390/app12136374) that can be mitigated by timed melatonin (doi: 10.1111/jpi.12730; doi:10.4103/1673-5374.332149).
Answer: Thank you for your suggestion. We have added the discussion of the correlation between melatonin and glucose tolerance in the discussion part. Please see Page 17 Line 453-461 in the revised manuscript R1.
Q5. Obviously, average plasma melatonin levels in chicks are much higher than in the mammalian species and in humans (lower than 5 pg/ml during daytime and about 10-fold higher by night, e.g., https://doi.org/10.1111/jpi.12657). Information on what values for melatonin parameters (Mesor, amplitude, phase) should be considered the reference values would be helpful to better comprehend the results of this study. Also, how much these values depend on age or gender?
Answer: Thank you for your suggestion. Regarding the parameters of melatonin in plasma, our study demonstrated that monochromatic light did not have an effect on the peak phase of melatonin, but had a significant effect on the median and amplitude of melatonin. Therefore, we believe that the median and amplitude of melatonin can respond to changes in monochromatic light. One study explored circadian rhythm changes in salivary melatonin levels during aging [1]. The decline in nocturnal peak levels (amplitude) in salivary melatonin was found in old and the oldest subjects. Both the old and the oldest subjects showed an increased daytime (baseline) melatonin level. Most strikingly, authors found that a step-wise decrease in the circadian rhythms of saliva melatonin occurred early in life, around 40 yr of ages. Compared to men, melatonin in women exhibits a significantly elevated amplitude rhythm [2], and pineal melatonin secretion was earlier relative to sleep time in women [3].
References:
- Zhou JN, Liu RY, van Heerikhuize J, Hofman MA, Swaab DF. Alterations in the circadian rhythm of salivary melatonin begin during middle-age. J Pineal Res 2003, 34(1), 11-6, doi: 10.1034/j.1600-079x.2003.01897.x.
- Gunn PJ, Middleton B, Davies SK, Revell VL, Skene DJ. Sex differences in the circadian profiles of melatonin and cortisol in plasma and urine matrices under constant routine conditions. Chronobiol Int 2016, 33(1), 39-50, doi: 10.3109/07420528.2015.1112396.
- Cain SW, Dennison CF, Zeitzer JM, Guzik AM, Khalsa SB, Santhi N, Schoen MW, Czeisler CA, Duffy JF. Sex differences in phase angle of entrainment and melatonin amplitude in humans. J Biol Rhythms 2010, 25(4), 288-96, doi: 10.1177/0748730410374943.
Q6. The authors speculate that melatonin may mediate effects of monochromatic light on the circadian rhythm of plasma insulin via the expression of clock gene in chicken pancreas. However, melatonin effect on circadian insulin production at least in mammals including humans is mainly attributed to MT1 and MT2 membrane receptors (see e.g., already refenced work#49, see as well doi: 10.1590/2359-3997000000098). Furthermore, it is known that mutation in MT2 receptor is linked to elevated fasting glucose (doi:10.1038/ng.290) along with prolonged melatonin secretion (doi: 10.2337/db15-0999) that may depend on the light perception (doi: 10.1111/jpi.12730). The authors do not at all mention melatonin receptors in Discussion, though, I believe, this issue deserves detailed discussion.
Answer: Thank you for your suggestion. In the discussion section, we have expanded on the discussion of melatonin receptors. We also included information on the distribution of melatonin receptors in the pancreas as well as the effect of melatonin receptors on insulin secretion. Please see Page 17 Line 467-476 in the revised manuscript R1.
MINOR
Q7. Material and methods. Providing information on light intensity (also) in lux rather than in W/m2 might be easier for readers to comprehend.
Answer: Thank you for your suggestion. We have changed the measure of light intensity to Lux in order to make it simpler for readers to comprehend. Please see Page 3 Line 105 in the revised manuscript R1.
Q8. Quality of figures should be increased as not all text with the statistical details is readable.
Answer: Thank you for your suggestion. The quality of all figures has been increased, including Figure 1-6 in the revised manuscript R1.
Reviewer 2 Report
Dear authors,
your paper is very interesting an can published after minor revision.
The linguistic style have to be improved by a native speaker.
The citated literature has to be checked - please check, if there are newer one.
Author Response
Dear Editor and Reviewers,
I’m very glad to hear from you, and thank you for your work in dealing with my manuscript. All of those comments are valuable and very helpful for improving the quality of our manuscript and the important guidance to our research. We have carefully checked the manuscript and revised it according to the suggestions. Now, I’m submitting here a reply to queries.
Reviewer #2:
Dear authors,
your paper is very interesting and can published after minor revision.
Q1. The linguistic style have to be improved by a native speaker.
Answer: Thank you for your suggestion. According to your suggestion, the grammar, punctuation, spelling, and vocabulary in this manuscript have been revised by a native speaker. Details are found in the marked modifications in the revised manuscript R1.
Q2. The citated literature has to be checked - please check, if there are newer one.
Answer: Thank you for your suggestion. The seven current references, which may be found in the newly listed literature and include 2,3,6,10,16,19 and 20, have been replaced.
Reviewer 3 Report
The manuscript by Song and colleagues provides an interesting insight into the relationship between circadian clock mechanisms and glucose metabolism.
It would be advantageous to provide more description of the circadian clock mechanism, how it works, and the roles of activators and suppressors – this would also provide bases for a needed explanation of why particular circadian clock genes were chosen.
Please provide a graphical scheme of the study, including time points, material, and exposures – it would make it easier for the reader to follow the manuscript, results in particular.
When genes appear for the first time in the manuscript, please provide the full name, similarly in the description of all figures. Other abbreviations used in the figures should also be expanded in the description of the figure.
The resolution and size of the figures should be increased.
Author Response
Dear Editor and Reviewers,
I’m very glad to hear from you, and thank you for your work in dealing with my manuscript. All of those comments are valuable and very helpful for improving the quality of our manuscript and the important guidance to our research. We have carefully checked the manuscript and revised it according to the suggestions. Now, I’m submitting here a reply to queries.
Reviewer #3:
The manuscript by Song and colleagues provides an interesting insight into the relationship between circadian clock mechanisms and glucose metabolism.
Q1.It would be advantageous to provide more description of the circadian clock mechanism, how it works, and the roles of activators and suppressors – this would also provide bases for a needed explanation of why particular circadian clock genes were chosen.
Answer: Thank you for your suggestion. We have described the mechanism of the circadian clock in the introduction part. Please see Page 2 Line 49-54 in the revised manuscript R1.
Q2.Please provide a graphical scheme of the study, including time points, material, and exposures – it would make it easier for the reader to follow the manuscript, results in particular.
Answer: Thank you for your suggestion. We have supplemented the schematic diagram of animal experiments in the revised Figure 1A. Please see Page 6 in the revised manuscript R1.
Q3.When genes appear for the first time in the manuscript, please provide the full name, similarly in the description of all figures. Other abbreviations used in the figures should also be expanded in the description of the figure.
Answer: Thank you for your suggestion. We have provided the full name of genes in the manuscript. Please see Page 2 Line 49-52, Page 4 Table1 and Page 7 Line 224 in the revised manuscript R1.
Q4.The resolution and size of the figures should be increased.
Answer: Thank you for your suggestion. The resolution and size of the figures have been increased, including Figure1-6 in the revised manuscript R1.
Reviewer 4 Report
The work of SONG and collaborators, "Role of melatonin in daily variations of plasma insulin level 2 and pancreatic clock gene expression in chick exposed to monochromatic light" aims to elucidate the effect of monochromatic light on circadian rhythms of plasma insulin level and pancreatic clock gene expression. To achieve this goal, they exposed hatched chicks (intact, sham, and pinealectomy) to white, red, green, or blue light for 21 days. Plasma melatonin and insulin concentrations and pancreatic clock gene expression were measured every 4 hours for circadian rhythm analysis. They report that red light decreased the mesor and amplitude of plasma melatonin decreased the mesor and amplitude of plasma melatonin and of pancreatic clock gene. In contrast, the red light increased plasma insulin concentration. After pinealectomy, the circadian expressions of plasma melatonin and the pancreatic clock gene were reduced, and as a result, the differences in insulin plasma in the red monochromatic light group were suppressed. In the light of these results, the authors conclude that melatonin and the clock gene in the pancreas regulate insulin secretion in the chicken pancreas.
The manuscript is well written, structured, and pleasant to read. This is a significant piece of work with great interest for readers interested in this field. The results are original, convincing, and significant. I have only minor remarks.
1/ Information on the expression of melatonin receptors (MT1 and/or MT2?) on beta cells should be added to the text and the diagram in figure 6.
2/I wonder about the implication of these results on the physiopathological aspects of diabetes and the metabolic syndrome. I wish the authors had discussed this point more.
3/ The legend for figure 1 is difficult to read (the order of the panels is a bit chaotic).
Author Response
Dear Editor and Reviewers,
I’m very glad to hear from you, and thank you for your work in dealing with my manuscript. All of those comments are valuable and very helpful for improving the quality of our manuscript and the important guidance to our research. We have carefully checked the manuscript and revised it according to the suggestions. Now, I’m submitting here a reply to queries.
Reviewer #4:
The work of SONG and collaborators, "Role of melatonin in daily variations of plasma insulin level 2 and pancreatic clock gene expression in chick exposed to monochromatic light" aims to elucidate the effect of monochromatic light on circadian rhythms of plasma insulin level and pancreatic clock gene expression. To achieve this goal, they exposed hatched chicks (intact, sham, and pinealectomy) to white, red, green, or blue light for 21 days. Plasma melatonin and insulin concentrations and pancreatic clock gene expression were measured every 4 hours for circadian rhythm analysis. They report that red light decreased the mesor and amplitude of plasma melatonin decreased the mesor and amplitude of plasma melatonin and of pancreatic clock gene. In contrast, the red light increased plasma insulin concentration. After pinealectomy, the circadian expressions of plasma melatonin and the pancreatic clock gene were reduced, and as a result, the differences in insulin plasma in the red monochromatic light group were suppressed. In the light of these results, the authors conclude that melatonin and the clock gene in the pancreas regulate insulin secretion in the chicken pancreas. The manuscript is well written, structured, and pleasant to read. This is a significant piece of work with great interest for readers interested in this field. The results are original, convincing, and significant. I have only minor remarks.
Q1. Information on the expression of melatonin receptors (MT1 and/or MT2?) on beta cells should be added to the text and the diagram in figure 6.
Answer: Thank you for your suggestion. The information on the expression of melatonin receptors on beta cells has be added to the text and the diagram in the revised Figure 6. Please see Page 17 Line 472-473 in the revised manuscript R1.
Q2. I wonder about the implication of these results on the physiopathological aspects of diabetes and the metabolic syndrome. I wish the authors had discussed this point more.
Answer: Thank you for your suggestion. According to your suggestion, We have added the effects of melatonin and the circadian clock on diabetes and metabolic syndrome in the discussion section. Please see Page 17 Line 446-453 in the revised manuscript R1.
Q3. The legend for figure 1 is difficult to read (the order of the panels is a bit chaotic).
Answer: Thank you for your suggestion. To make it easier to read, we have rearranged the panels in figure 1 and revised the legend interpretation. Please see Page 7 Line 208-222 in the revised manuscript R1.
Round 2
Reviewer 1 Report
The authors fully addressed the main issues and improved manuscript. I can recommend it for publication.